# Service-policy gaps in the settlement journey of Arabic-speaking immigrant newcomer and refugee older adults in Edmonton, Canada

**Saba Nisa**●*●, **Sadaf Murad**●●, **Jordana Salma**●●, **Alesia Au**●●

Faculty of Nursing, University of Alberta, Edmonton, Canada

● These authors contributed equally to this work.
* snisa@ualberta.ca

**Data Availability Statement:** There are ethical and legal restrictions imposed by the University of Alberta which prevent the public sharing of minimal

## Abstract

Immigrant newcomers and refugees (INRs) are two migrant categories that experience consistent systemic barriers to settlement and integration in Canada as older adults. This paper explores the challenges experienced by Arabic-speaking INR older adults in Edmonton, Canada, during settlement and discusses policy and service implications. A qualitative description study using community-based participatory research principles was implemented to evaluate and support digital literacy in Arabic-speaking INR older adults. We included men and women aged 55 and older who identified as immigrants or refugees and spoke Arabic. Experiences of settlement challenges were consistently identified during data collection and engagement of INR older adult participants. A thematic sub-analysis of interviews with (10 individuals and one couple) of participants' narratives was completed in 2022 and was used to identify themes related to settlement barriers for this population. Two main themes were identified: (1) Limited English skills and digital literacy gaps create service barriers for INR older adults, and (2) Gaps in services and policies as basic needs remain unmet. We describe key challenges experienced by INR older adults, such as language barriers, precarious finances, poor access to health care services and lack of transportation and employment opportunities, which hinder successful integration into the new society. This study showcases the ongoing challenges with early settlement and integration that continue despite Canada's well-developed immigration settlement landscape. INR older adults often remain invisible in policy, and understanding their experiences is a first step to addressing their needs for resources that support healthy aging in the post-migration context.

## Introduction

Canada has welcomed immigrant newcomers and refugees (INRs) for many years. An immigrant in Canada has the official right, granted by immigration authorities, to live in the country permanently as a landed immigrant or permanent resident [1]. According to the Government of Canada [2], refugees are individuals who have fled their countries because of a well-founded

data for this study due to patient confidentiality concerns. Data are available upon request from University of Alberta, Research Ethics Office, via email (safegrd@ualberta.ca) for researchers who meet the criteria for access to confidential data.

**Funding:** The author(s) received no specific funding for this work.

**Competing interests:** The authors have declared that no competing interests exist.

fear of persecution. Newcomers refer to individuals who immigrated to Canada for work, school, or to live within the past five years [3]. Compared with the other G7 nations (France, Germany, Italy, Japan, the UK, and the US), Canada had the highest growth rate of its immigrant population between 2018 and 2019 and resettled 30,087 refugees [4] and over 1.3 million new immigrants settled permanently in Canada from 2016 to 2021, the highest number of recent immigrants recorded in a Canadian census [5]. By 2036, immigrants will represent up to 30% of Canada's population, including refugees [5]. Immigration accounts for almost 100% of Canada's labour force growth [5]. Roughly 75% of Canada's population growth comes from immigration, mainly in the economic category [5].

Despite contributing to Canada's economy and community, immigrant newcomers and refugees encounter several challenges when beginning their lives in Canada that impact their livelihoods and health status [3]. Compared with other immigrant groups who may choose to leave their country, refugees are not emigrating for economic reasons but rather to seek protection from social and political insecurity [6]. After one year of government financial support, refugees are expected to become self-sufficient, which involves seeking and finding work. Immigrant newcomers, on the other hand, have even less support and are expected to be financially self-sufficient on arrival to Canada. Yet, economic integration has consistently been challenging for all immigrants, especially refugees [7]. The wellness of INR is negatively impacted by work instability, inadequate housing, minimum wages and poor work conditions, the lack of access to professional training equivalency and accreditation, and difficulties with language acquisition [8]. Migrating after retirement often translates into dependency on family and loss of autonomy [9]. Family-sponsored immigrants and refugees might not qualify for all types of government-funded social and health services which further exacerbates settlement challenges in older age [10]. As the population ages and the proportion of INRs increases, federal and provincial policymakers in Canada face challenges in adequately addressing the diversified needs of INR older adults [11].

The Arabic-speaking INR population is growing fast. Over 480,000 Arabic-speaking immigrants live in Canada (Statistics Canada, 2017). Arabic-speaking INRs to Canada originate from at least 20 different countries [12]. They share many beliefs, traditions, and values, particularly concerning the care and support of older persons that includes extended families living together for constant support and care needs [13,14]. Moreover, maintaining cultural and religious practices is important for older Arabic-speaking immigrants but can be challenging in a new country [15]. These individuals often face difficulties accessing culturally relevant community services, such as places of worship, halal food, and culturally specific activities [15]. Adjusting to Canadian cultural norms while trying to preserve their own traditions can create stress and identity conflicts for older adults [16].

Some studies have identified that INR older adults experience challenges during settlement and integration, such as language barriers, social isolation, distrust of social services and financial precocity [17,18]. However, less attention has been paid to examining the lived experiences of Arabic-speaking older adult most at risk for barriers to accessing services and supports. This qualitative descriptive study aims to examine the experiences of Arabic-speaking INR older adults' during their settlement and integration process in Edmonton, Canada, and explore the gaps in services and policies.

## Methodology

We used a qualitative descriptive study design to gain insights into the settlement experiences of Arabic-speaking INRs in Edmonton as part of a larger community-based participatory research project aimed at addressing digital literacy in this population. Qualitative description allows for a thorough exploration of the phenomenon, enabling a comprehensive

understanding of this population's specific challenges and a platform for amplifying the voices of individuals experiencing the phenomenon of interest [19,20]. This study received ethics approval from the human research ethics board of the first author's institution (University of Alberta). Informed written consent was taken from the participants.

### Inclusion/Exclusion criteria

Participants were included in the sub-study if they met the following inclusion criteria: 1) 55 years of age or older; 2) self-identification as an Arabic-speaking immigrant newcomer or refugee. A social service organization that caters to Arabic-speaking migrants supported the recruitment of participants who were part of their clientele base. The research team conducted convenience sampling by contacting the organization's clients via phone to explain the research study. The researcher received written consent from participants to communicate the research's objectives, procedures, potential risks, benefits, and the voluntary nature of participation. All potential participants who met the inclusion criteria were invited to participate in the study. The majority were women, younger than 75 years of age and self-identified as having low income and limited fluency in English. Among the participants (four participants) were receiving some form of financial support from the Canadian Government. The participants varied in terms of education level and length of time in Canada, with the majority being in Canada for less than five years—characteristics of details for participants (Table 1).

### Data collection

Data included in this sub-study stemmed from 10 semi-structured individual interviews and one interview with a couple. Socio-demographic data were collected using a brief questionnaire before the interview. An interview guide was utilized to explore their use of digital technology and learning needs. Settlement challenges emerged organically in conversation and were probed further by the research team. Three interviews were conducted in English and eight were conducted in Arabic language. The interviews lasted 1–2 hours and were completed from April to June 2022. All interviews were led by a graduate nursing student experienced in qualitative interviewing and an Arabic interpreter with lived experience in the community. Participants were invited to meet the researchers and participate in the interview at a convenient place, such as their home or the social service organization. One interview was completed over the phone as the participant was concerned about COVID-19 transmission. Field notes and reflexive memos guided data collection.

### Data analysis

All interviews and field notes were thematically analyzed using NVivo 12 qualitative data analysis software. Braun and Clarke's (2006) [20] six-phase thematic analysis model was used and involved becoming familiar with the data, coding to capture the semantic and conceptual meaning of the data, constructing themes through inductive and deductive processes, and then refining and naming themes. Study codes were compared and discussed in team meetings. We searched for variations in experiences among participants based on their gender, ethnicity, socioeconomic status, English language ability, and time since migration.

### Trustworthiness

We aimed to ensure rigour by establishing philosophical coherence, demonstrating clarity of intent, and maintaining integrity within the methodology, processes, and outcomes of the study.

**Table 1. Characteristics details of participants.**

| Characteristics of participants | n |
|---|---|
| **Gender** | |
| Female | 8 |
| Male | 4 |
| **Age (Years)** | |
| 50–60 | 7 |
| 61–70 | 3 |
| 71–85 | 2 |
| **Country of origin** | |
| Middle East (Lebanon, Palestine, Iran and Syria) | 11 |
| Africa (Somalia, Uganda, Kenya, Algeria, and Egypt) | 1 |
| **Immigration Status** | |
| Canadian Status | 3 |
| Family sponsored | 1 |
| PR with Government sponsored refugee | 7 |
| Privately Sponsored | 1 |
| **Time in Canada (years)** | |
| $\geq$10 | 2 |
| Between 5–10 | 2 |
| $\leq$5 | 8 |
| **Level of education** | |
| Elementary | - |
| Secondary | 5 |
| Postsecondary | 7 |
| **Self-Related English fluency** | |
| Minimal to none | 8 |
| Average | - |
| Good | 3 |
| Excellent | 1 |
| **Household income (per year)** | |
| $\leq$\$20,000 (within low-income bracket for province) | 8 |
| Between \$20,000–\$40,000 | 3 |
| $\geq$\$40,000 | 1 |
| **Source of Income** | |
| Governmental Welfare | 4 |
| Job (part or full time) | 4 |
| Family support | 2 |
| Retirement | 1 |
| Unknown | 1 |

The principal investigator's (last author) positionality as an Arabic immigrant woman from Lebanon facilitated rapport-building with participants, enhancing the study's credibility. To mitigate personal bias, all team members, including research assistants, maintained reflexive notes and audit trails after interviews. To mitigate bias, an inductive analysis method was employed, and multiple researchers (XX, XX, XX) independently reviewed the transcribed interviews and memos. To enhance confirmability, we engaged in reflexive journaling, documenting our thoughts, decisions, and potential biases during the research process. Weekly debriefings among the team provided essential support, validated findings, and further

minimized bias, ensuring the dependability and confirmability of the study outcomes. Codes, subcategories, and categories were iteratively discussed until a consensus was reached. Additionally, we scrutinized the data for alternative interpretations with the whole team, which strengthened our findings.

## Findings

The following themes highlight the challenges faced by Arab-speaking INR older adults who, despite receiving some support from a local social service agency, identified significant ongoing gaps in support due to systemic barriers that impacted their well-being (Fig 1).

### 1 Limited English skills and digital literacy gaps create service barriers

The study's findings highlight several interrelated challenges faced by INR. Navigating resources proves complex due to unfamiliar systems and limited formal social support networks. Language barriers, specifically limited English skills, hinder effective communication and resource utilization. Participants faced complex challenges in navigating resources due to limited language skills and inadequate formal support at both personal and organizational levels. Many struggled with finding doctors, accessing ethnic foods, and obtaining government assistance. This lack of formal support to access basic necessities adversely affected the well-being of older adults, highlighting the need for improved support systems.

"You know, there are things that are sensitive, like halal meat. We just needed someone to guide us, but it was difficult. The one supposed to do that was in Saudi Arabia visiting her husband, and nobody was there to show us around" (Participant 1).

Participants shared a disparity between newcomers' initial expectations of comprehensive assistance and the reality of needing to actively seek and manage information and resources on their own to navigate their new surroundings effectively. Often case workers are assigned for a specific period of time early in settlement but older adults reported requiring ongoing support after this initial period of support and which went beyond the mandate of some of the settlement organizations.

"When you come here as a newcomer, you're expecting some organizations to supplement you with many things. Well, they don't. And they try their best, but you find yourself—like, you have to research on your own for everything. In my country, if I want to have a ride and I go to a mosque, for example, using the transit, I don't need to research. Here you have to research all the time. Like time and Google, a lot of those things. This is another thing that I think people who are coming here, mostly they are not used to that" (Participant 12).

Some participants pointed out that when seniors visit government offices for assistance, they are often directed to websites to find information. This approach poses challenges, particularly for older adults with combined computer literacy and language barriers. One participant emphasized that this situation adds a layer of difficulty to settlement for older adults who do not have family locally or a strong social support network and who might also have barriers to leaving their homes to access in-person supports due to mobility and transportation barriers.

"Even here, when you go to a government office to do anything, for example, they give you a website and ask you to see for yourself; and this can be hard for seniors, who, other than the computer literacy, already have a language problem" (Participant 12).

Codes

*Lack of language skills (14)

*No support to access services (4)

*Unable to navigate services because of limited language skills (13)

*Financial challenges (10)

*Unable to afford groceries; food cost too high (9)

*Transportation issues (2)

*Waiting for subsidized housing (4)

**Subthemes**

*Lack of formal social support

*Limited English language skills

**Subthemes**

*Housing challenges

*Limited access to food

*Health vulnerabilities

**Theme 1**

Limited English Skills and Digital Literacy Gaps Create Service Barriers

**Theme 2**

Gaps in Services and Polices as Basic Needs Remain Unmet

**Fig 1. Presentation of qualitative findings by themes, subthemes and codes.**

Language learning programs were a strong source of support by helping with information access and socialization to Canada. Some participants were dissatisfied with language-learning programs, indicating that most English courses for immigrants fail to meet the specific needs of older adults, such as support for digital literacy, accommodation of disabilities, and recognition of other social obligations that older adults might have within their families, such as caregiving. This mismatch prevents effective language enhancement, with some participants repeating lower-level courses without genuine educational commitment or quitting altogether.

"I made it to level 6, and we needed to start writing. I struggled to write, so I asked them to give me a chance. My grades were not good in writing: "Give me a chance to redo the course." They refused and said, "No, you can't do this." However, there are people who are stuck in levels 3, 4, and 1 and 2; they're repeating the levels three or four times" (Participant 12).

Therefore, despite existing services and supports in place, participants experienced challenges in navigating these resources when they needed them, leading to experiences of dissatisfaction and frustration.

## 2. Gaps in services and policies as basic needs remain unmet

Another primary concern highlighted in the interview data was financial challenges. Older adult INR participants struggled with the necessities of life and housing issues, often related to their limited language competency and unemployment. The participants worked with inadequate income and finances to support themselves and their families. Some participants reported that their age and health limitations made it challenging to find jobs, exacerbated their financial constraints, and impacted their quality of life. A few participants faced significant financial challenges related to paying their rent, with some on long waiting lists for subsidized housing. This participant reflects on a more comfortable life in Syria, highlighting the contrast between their past and current living conditions.

"Here, rent in Canada is a huge problem. When it comes to clothing, I never buy new clothes; I go to Value Village, and I buy clothes, but I tell myself that it's not a big deal. However, renting places can be expensive—$1,300. I pay $995 towards rent, you see? Of course, they cover water and heat, and I pay electricity, and Internet every month costs about $200. I also buy medication every month. I mean, we came here, but to be honest, we were living a much more comfortable life in Syria" (Participant 13).

Some older adult INRs depended on social service organizations and food banks for food because their limited finances did not allow them to buy groceries independently. For most of the older adults, their low socioeconomic status increased their fear of food insecurity for themselves while sometimes needing to support family left-behind in countries of origin. Providing financial remittances in times of difficulty to family outside of Canada was still an expectation participants had for themselves.

"I do, but they are struggling more than me you know? With the Lebanon situation, no water, no power, no heat, no jobs, no—we're still a lot better. We have a food bank. They don't. You know, we struggle but we [are] 100 percent better than them. You got, you know, food bank. You got, you know, a pension. They don't. Nothing, you know. And I wish I could help them, and how could I be short? But I'm still a lot better than Lebanese. How do they survive today? . . . I'm eating, you know, from community organization and I got some meat, I got some chicken, you know. Only I have to buy some vegetables or—

grains they provide us. Sometimes, I cook what they got me for a whole week, and thanks to them, it helps" (Participant 10)

Health is another major area in which older adult INRs struggled due to limited finances and language skills. The participants reported that access to healthcare services was challenging, negatively impacting their physical and mental health. Financial constraints limited their ability to pay out-of-pocket for medications and other healthcare services:

"My hand, I can't move it up. I did a few physio sessions. It's hard without an interpreter, so it's not being corrected properly. If they speak with the Government, they could increase the money and support me with this, but I need an interpreter and can't get one. I am struggling. I can't even sleep at night (Participant 01).

Being able to find affordable places to live and have financial security was of upmost importance to the participants. The lack of financial resources to support their livelihood was detrimental to their health and wellbeing.

## Discussion

The challenges of INR older adults cannot be viewed as short-term ones that can be resolved exclusively by means of 'exceptional' or 'emergency' responses. Violence, political oppression, human rights abuses, and desires by people for a better life and greater economic opportunity will continue to act as sources of migration. It is therefore important to search for solutions that recognize older adult INRs as essential to the wellbeing of our communities and ensure structural and policy changes accordingly [21].

Accessing services and settling in Canada pose multiple challenges for older adult INRs from Arabic- Speaking communities and this is mirrored in other studies on INR populations [7,21–26]. Participants in our study, despite being connected to a social service organization, reported insufficient support in several areas. They faced difficulties with navigating services and resources, encountered language barriers, and struggled with housing and food insecurity. Concerns about food costs leads to reduced diet quality and quantity, and involuntary cuts in food intake. This situation can result in hunger and have serious negative effects on both physical and mental health [27]. Such challenges forced some participants to take low-paying jobs and withdraw from English-language classes so that they could meet their basic needs. Our findings also demonstrate that English language proficiency is a pivotal challenge affecting older adults' settlement experiences. This limitation is interlinked with other factors, such as access to employment and healthcare services, financial security, and social engagement in the community. These findings mirror work by others who identify language as a major barrier to settlement in Canada [14,26]. Other major barriers reported by participants were scarcity of culturally appropriate services and limited understanding by service providers of the needs of older migrants. Many conventional support services may not accommodate specific dietary restrictions, such as halal food requirements, which can be crucial for maintaining their religious practices. Additionally, the lack of culturally sensitive approaches in healthcare and social services can lead to a feeling of alienation or discomfort. These barriers can prevent older adults from seeking necessary support, impacting their overall well-being [28]. Furthermore, these challenges were exacerbated by difficulties associated with transportation, which prevent older adults from participating in community programs and gaining access to the resources that are available to them [29].

Programs and initiatives that support older adults are often different from those that support immigrants and refugees and involve different ministries, organizations, and levels of government, which has resulted in a lack of cohesion among formal supports across sectors [15]. Furthermore, INR older adults might not be aware of available health and social services in their communities, which can negatively impact their health [14,22,25,26]. Key recommendations to address the above-described gaps in services and supports include increasing information access, government income support programs, and language classes specific for older adults, and accessible housing options [30].

## Strengths and limitations

The strength of our study is that we included the perspectives of immigrants who came for different reasons (family-sponsored, economic and refugees) that contextualized the migration context in Canada. Reasons for migration create differences in migration experiences. Another strength is that there is a lack of research that can contextualize the intersections of challenges that Arabic IRNs face, and so this qualitative work allows for their voices to be heard. Moreover, the individual/group discussion was important for family members to engage in research, but also the fact that Arabic interpretation was provided to ensure that language barriers did not limit data collection. The limitation of this study is that we lacked perspectives of participants who are family-sponsored (only one participant). Therefore, these findings may not reflect the perspectives of those who live in multigenerational homes and have more active, local family support.

## Conclusion

This study offers important insights into the settlement challenges faced by older Arabic-speaking immigrants and refugees in Canada. It highlights gaps in support and services and examines the difficulties of balancing cultural and religious traditions with the demands of adapting to a new society. The findings underscore the need for targeted support services to address the unique needs of this group and aid their integration into Canadian society.

## Author Contributions

**Conceptualization:** Jordana Salma, Alesia Au.

**Data curation:** Alesia Au.

**Formal analysis:** Saba Nisa, Sadaf Murad, Jordana Salma.

**Writing – original draft:** Saba Nisa, Sadaf Murad, Jordana Salma.

**Writing – review & editing:** Saba Nisa, Sadaf Murad, Jordana Salma.

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
