## [Decision Letter · Decision Letter 0]

8 Aug 2024

PONE-D-24-08291Service-Policy Gaps in the Settlement Journey of Arabic-speaking Immigrant Newcomer and Refugee Older Adults in CanadaPLOS ONE

Dear Dr. Nisa,

Thank you for submitting your manuscript to PLOS ONE. After careful consideration, we feel that it has merit but does not fully meet PLOS ONE’s publication criteria as it currently stands. Therefore, we invite you to submit a revised version of the manuscript that addresses the points raised during the review process. This is an interesting study that adds to the current literature. In your revision, please address "how trustworthiness of the data was assured, i.e., credibility, transferability, dependability and confirmability" (per Reviewer 1). Also, please make additional clarifications and edits based on all feedback provided, including from Reviewer 2.

We look forward to receiving your revised manuscript.

Kind regards,

Magdalena Szaflarski, PhD

Academic Editor

PLOS ONE

A clean copy of the edited manuscript (uploaded as the new *manuscript* file

Reviewers' comments:

Reviewer's Responses to Questions

**Comments to the Author**

1. Is the manuscript technically sound, and do the data support the conclusions?

Reviewer #1: Yes

Reviewer #2: Yes

2. Has the statistical analysis been performed appropriately and rigorously? 

Reviewer #1: Yes

Reviewer #2: N/A

3. Have the authors made all data underlying the findings in their manuscript fully available?

Reviewer #1: Yes

Reviewer #2: Yes

4. Is the manuscript presented in an intelligible fashion and written in standard English?

Reviewer #1: Yes

Reviewer #2: Yes

5. Review Comments to the Author

Reviewer #1: Would liked to have seen how trustworthiness of the data was assured, i.e., credibility, transferability, dependability and confirmability. Describing these methods will strengthen the paper. Overall and interesting paper highlighting important concerns of older Arabic speaking INRs in Canada. I was under the wrong impression that Canada had less resettlement challenges than the US.

Reviewer #2: This interesting paper presents an important qualitative study exploring the challenges experienced by Arabic-speaking immigrant/refugee older adults in Edmonton, Canada during settlement. The paper discusses policy and service implications. The topic of the study will be of value for public health specialists and health/social care providers. Congratulations to the authors. I am offering some suggestions for consideration before publication.

Title

1. Please specify the location; that the study was among Arabic-speaking immigrant/refugee older adults in Edmonton, Canada

Abstract

2. Like comment above, kindly specify location (i.e., Edmonton). This will help the reader gain insight into the exact location/province in Canada where the study was conducted.

3. Please briefly mention eligibility criteria.

Introduction

1. Minor typo: in sentence “…. to live within the past five years [3],” please replace comma with period before the following sentence.

2. For sentence “It resettled 30,087 refugees [4].” Do the author mean between 2018-2019? If so, please connect to the previous sentence.

3. The authors state “Just over 1.3 million new immigrants settled permanently in Canada from 2016 to 2021….”. Does this number include refugees?

4. Same question for “By 2036, immigrants will represent up to 30% of Canada's population (5)”. Does this number involve refugees?. Also, kindly change the reference from (5) to [5] to be consistent with reference style used.

5. For “Immigration accounts for almost 100% of Canada's labour force growth”, please add reference.

6. The authors state that “They share many beliefs, traditions, and values, particularly concerning the care and support of older persons”. Please give a few examples of these traditions, values.

7. It is thoughtful that the authors mention some of the challenges faced by INR older adults (e.g., language barriers, social isolation, distrust of social services and financial precocity). It would be insightful to please add some of the distinct barriers that Arabic speaking INR older adults may face to highlight the importance of conducting this study with this population.

Methods

8. By “the researcher team”, do the authors mean “the research team”? If so, please revise for clarity.

9. The authors state “The majority were women, younger than 75 years of age and self-identified as having low income and limited fluency in English. Most were receiving some form of financial support from the Canadian Government”. If possible, please briefly give specific numbers/percentage so the reader can gain some insight when reading the text.

10. Thank you for outlining the details about the interviews. As per methods section, I understand that 11 interviews (10 individual and 1 with couple) were conducted. Would you please check the abstract and clarify/address any typo in this regard?

11. Would you please give a brief insight into how many interviews were done in Arabic?

Discussion

12. Please briefly discuss findings on food insecurity and how this aligns with findings from other Canadian research, as this is a significant problem in Canada. You may find this recent study in Ontario, Canada helpful (https://doi.org/10.1016/j.appet.2024.107226).

13. The authors state “Other major barriers reported by participants were scarcity of culturally appropriate services and limited understanding by service providers of the needs of older migrants.”. Please briefly discuss this important finding, as lack of culturally sensitive services is a significant problem for Arabic speaking immigrants/refugees. You may find this paper helpful

https://doi.org/10.1007/s00127-024-02668-4.

14. Please add a brief strengths/limitations section for your thoughtful paper.

15. Please add a brief conclusion section.

6. PLOS authors have the option to publish the peer review history of their article (what does this mean?). If published, this will include your full peer review and any attached files.

Reviewer #1: No

Reviewer #2: No

---

## [Author Response · Author response to Decision Letter 0]

28 Aug 2024

Dear Editor and Reviewers,

Thank you for your time and efforts in reviewing our manuscript. We greatly appreciate your feedback, which has helped enhance the readability and quality of the paper. We have addressed all of your comments and made the necessary revisions.

Thank you once again for your valuable input.

---

## [Editor Report · Decision Letter 1]

4 Sep 2024

PONE-D-24-08291R1Service-Policy Gaps in the Settlement Journey of Arabic-speaking Immigrant Newcomer and Refugee Older Adults in Edmonton, CanadaPLOS ONE

Dear Dr. Nisa,

Thank you for submitting your manuscript to PLOS ONE. After careful consideration, we feel that it has merit but does not fully meet PLOS ONE’s publication criteria as it currently stands. Therefore, we invite you to submit a revised version of the manuscript that addresses the points raised during the review process.

Most critiques have been addressed in this revised manuscript. However, it is not clear how the revision addresses the issue of trustworthiness of the data (credibility, transferability, dependability, and confirmability; per Reviewer 1). Please provide an explanation and point to specific part/s of the manuscript where this information is presented.

We look forward to receiving your revised manuscript.

Kind regards,

Magdalena Szaflarski, PhD

Academic Editor

PLOS ONE
---

## [Author Response · Author response to Decision Letter 1]

19 Sep 2024

Dear Reviewer,

Thank you so much for your feedback on the manuscript. We have addressed and made the component of trustworthiness clear. Your comments are valuable in enhancing the quality of our paper.

---

## [Editor Report · Decision Letter 2]

24 Sep 2024

Service-Policy Gaps in the Settlement Journey of Arabic-speaking Immigrant Newcomer and Refugee Older Adults in Edmonton, Canada

PONE-D-24-08291R2

Dear Dr. Nisa,

We’re pleased to inform you that your manuscript has been judged scientifically suitable for publication and will be formally accepted for publication once it meets all outstanding technical requirements.

Kind regards,

Magdalena Szaflarski, PhD

Academic Editor

PLOS ONE
---

## [Editor Report · Acceptance letter]

10 Oct 2024

PONE-D-24-08291R2 

PLOS ONE

Dear Dr. Nisa, 

I'm pleased to inform you that your manuscript has been deemed suitable for publication in PLOS ONE. Congratulations! Your manuscript is now being handed over to our production team.

Kind regards, 

on behalf of

Dr. Magdalena Szaflarski 

Academic Editor

PLOS ONE